# Exploring the Use of a Modified High-Temperature, Short-Time Continuous Heat Exchanger with Extended Holding Time (HTST-EHT) for Thermal Inactivation of Trypsin Following Selective Enzymatic Hydrolysis of the β-Lactoglobulin Fraction in Whey Protein Isolate

**DOI:** 10.3390/foods8090367

**Published:** 2019-08-26

**Authors:** Laura Sáez, Eoin Murphy, Richard J. FitzGerald, Phil Kelly

**Affiliations:** 1Teagasc Food Research Centre, Moorepark, Fermoy, P61 C996 Co. Cork, Ireland; 2Department of Biological Sciences, University of Limerick, V94 T9PX Limerick, Ireland; 3Food for Health Ireland, Teagasc Food Research Centre, Moorepark, Fermoy, P61 C996 Co. Cork, Ireland

**Keywords:** Hydrolysis, WPI, trypsin, α-lactalbumin, β-lactoglobulin, thermal inactivation

## Abstract

Tryptic hydrolysis of whey protein isolate under specific incubation conditions including a relatively high enzyme:substrate (E:S) ratio of 1:10 is known to preferentially hydrolyse β-lactoglobulin (β-LG), while retaining the other major whey protein fraction, i.e., α-lactalbumin (α-LA) mainly intact. An objective of the present work was to explore the effects of reducing E:S (1:10, 1:30, 1:50, 1:100) on the selective hydrolysis of β-LG by trypsin at pH 8.5 and 25 °C in a 5% (*w*/*v*) WPI solution during incubation periods ranging from 1 to 7 h. In addition, the use of a pilot-scale continuous high-temperature, short-time (HTST) heat exchanger with an extended holding time (EHT) of 5 min as a means of inactivating trypsin to terminate hydrolysis was compared with laboratory-based acidification to <pH 3 by the addition of HCl, and batch sample heating in a water bath at 85 °C. An E:S of 1:10 resulted in 100% and 30% of β-LG and α-LA hydrolysis, respectively, after 3 h, while an E:S reduction to 1:30 and 1:50 led >90% β-LG hydrolysis after respective incubation periods of 4 and 6 h, with <5% hydrolysis of α-LA in the case of 1:50. Continuous HTST-EHT treatment was shown to be an effective inactivation process allowing for the maintenance of substrate selectivity. However, HTST-EHT heating resulted in protein aggregation, which negatively impacts the downstream recovery of intact α-LA. An optimum E:S was determined to be 1:50, with an incubation time ranging from 3 h to 7 h leading to 90% β-LG hydrolysis and minimal degradation of α-LA. Alternative batch heating by means of a water bath to inactivate trypsin caused considerable digestion of α-LA, while acidification to <pH 3.0 restricted subsequent functional applications of the protein.

## 1. Introduction

Protein hydrolysis is a well-established process whereby proteins are cleaved to form peptides of different sizes under aqueous conditions. The nature of the peptides obtained depends upon the site(s) of protein cleavage, which in turn defines their length. Hydrolysis may be undertaken using alkali, acid or enzymes. An advantage of enzymatic hydrolysis is that the process usually takes place under relatively mild operating conditions—in contrast to alkali or acid hydrolysis, where extreme pH and temperature values may affect protein structure [1,2].

Enzymatic hydrolysis is commonly applied in many fields, i.e., waste treatment [3,4,5], detergent formulation [6] and at various stages during food processing. The products and ingredients derived from the hydrolysis of proteins in the food industry are of particular significance due to their biofunctional properties [7]. This is in addition to the enhanced technofunctional properties obtainable following enzymatic hydrolysis, e.g., enhanced solubility, foaming capacity, emulsion and gelation [8,9,10,11,12,13]. Other potential applications of enzymatic hydrolysis include the reutilisation of byproduct streams of food processing previously considered waste and a problem in terms of negative environmental impact [14,15]. A further field of interest for the food industry is the characterisation of the nutritional and bioactivity properties of peptides generated by enzymatic hydrolysis, which in recent years has highlighted potential biomarkers for physiological benefit such as antioxidant activity, regulation of gastric transit, antimicrobial activity, anticaries activity, antihypertensive activity, anti-inflammatory, satiety control and reduction of allergenic potential [7,16,17,18,19,20,21,22,23]. Reducing allergenic potential by enzymatic hydrolysis is especially important in the case of the dairy industry. Cow’s milk allergy (CMA) is considered the most common allergy in children under three years of age. The reaction is triggered by one or more of the milk proteins, the most allergenic of which are represented in the following order: caseins, β-LG, α-LA [24]. Due to the heat-labile nature of whey proteins, their allergenicity may be modulated to some extent by certain thermal processing treatments, thus making their use in food formulations, in the form of whey protein concentrate (WPC) or whey protein isolate (WPI), preferential compared to the more heat-stable caseins. It is notable, however, that the β-LG fraction, which represents approximately 53% of the whey protein in cow’s milk, is not present in human milk.

Enzyme activity ideally requires a combination of “optimum” conditions of temperature, pH, concentration of protein substrate and enzyme type [25,26,27,28] in order to maximise the degree of hydrolysis (DH or %DH). Thus, the optimum point of hydrolytic activity represents the set of conditions under which the highest conversion is achieved in a given time. Much work has focussed on maximising %DH in order to produce more and shorter peptides; however, there is increasing appreciation that intermediate %DH levels may have a more pronounced influence on final hydrolysate properties [29,30]. Furthermore, numerous substrate pre- or post-treatments have also been investigated in order to expand potential hydrolysate functionality [31], e.g., thermal pre-treatments [32], high pressure treatments [33] or a combination of endo- and exo-proteases [34].

Recent studies have shown that targeted or selective hydrolysis of proteins generates hydrolysates with specific peptides that have different properties to those of a fully hydrolysed protein [35,36,37,38,39,40]. The operating protocol necessary to achieve selective protein hydrolysis demands a specific set of incubation conditions, which may be less than optimum for the hydrolysis reaction per se [40]. In the case of WPI, a selective hydrolysis of β-LG may reduce the risk of allergy and at the same time enable the generation of a co-product enriched in α-LA—the predominant whey protein fraction in human milk, which is a key nutritional source of essential amino acids in sufficient concentration to promote infant development. Supplementation of infant formula with α-LA helps lower the total protein content of formula to resemble that of human milk. In addition, α-LA possesses a wide range of biological activities such as absorption of minerals, antibacterial activity, antioxidant activity, immunomodulatory effects and antitumor activity, which promote the health of the neonate [41,42]. The specific conditions reported for the selective hydrolysis of β-LG by trypsin or chymotrypsin include temperature (25 °C) and pH (pH 8.5), which resulted in minimal hydrolysis of the α-LA present [43,44,45].

At an industrial scale, thermal processes are frequently used to inactivate enzymes at the end of a reaction or on attaining a desired %DH. Lisak et al. [45] used non-thermal methods of inactivating chymotrypsin following WPI-based selective digestion of β-LG in order to avoid the risk of enzyme reactivation during time lags in thermal transfer buildup to the target temperature, resulting in excessive digestion of the major WPI fractions. Hence, the addition of inhibitors and acidification to <pH 3 were employed as alternative methods of enzyme inactivation. There are practical constraints, however, when using such approaches, i.e., cost, in the first instance; while heavily acidified substrates in the latter require neutralisation before further processing, thereby also potentially leading to enzyme reactivation.

In this study, a continuous high-temperature, short-time heater (HTST) with extended holding time (EHT)-based thermal treatment was assessed as a means of inactivating trypsin-based selective hydrolysis of β-LG in a WPI substrate dispersion while quantifying the formation of undesirable side reactions associated with protein denaturation and protein‒peptide aggregation. The ultimate objective was to minimise possible aggregate formation that could interfere with subsequent research efforts aimed at producing an α-LA-enriched product from the targeted WPI hydrolysis. Furthermore, since the original research [43] using trypsin was conducted at a relatively high E:S of 1:10, a further objective was to evaluate a reduced E:S (down to 1:100) in terms of the incubation conditions necessary to maximise β-LG digestion while minimising the digestion of α-LA.

## 2. Material and Methods

### 2.1. Targeted Enzymatic Hydrolysis

Commercially sourced whey protein isolate (WPI) Isolac^®^, provided by Carbery Food Ingredients, Ballineen, Ireland, with a total protein content of 91.4% was dispersed at 5% (*w*/*v*) in distilled water using an electronic overhead stirrer model VWR VOS 40 digital (VWR International Ltd., Dublin, Ireland). The three-bladed propeller (55 mm in diameter) stirrer was set at 300 rpm for 30 min at room temperature during WPI reconstitution and the dispersions were allowed to stand overnight at 4 °C to allow complete hydration.

Enzymatic proteolysis was carried out using a commercial trypsin preparation (Trypsin 250, a porcine pancreas-based extract supplied by Biocatalysts, Cardiff, UK) with a specific activity of 250 U/mg protein. Incubation conditions were set at 25 °C and pH 8.5 in accordance with the conditions described [43] in order to pursue targeted hydrolysis of β-LG in the WPI dispersion. In order to further optimise this targeted enzymatic hydrolysis, samples were prepared at different enzyme:substrate ratios (E:S) (1:10, 1:30, 1:50, 1:100) (*w*:*w*) and over hydrolysis durations ranging from 1 to 7 h. Hydrolysate samples (5 mL) were taken hourly and the reaction was stopped by the addition of 1 mL of 1 M HCl in order to reduce to pH < 3 according to the enzyme supplier’s protocol. The samples were frozen at −20 °C and retained for further analysis.

The enzymatic reaction was controlled using a Metrohm 842 Titrando pH-STAT titrator (Herisau, Switzerland), which maintained pH 8.5 by dosing 1M NaOH throughout the hydrolysis period.

### 2.2. Degree of Hydrolysis (%DH)

Quantification of %DH by the pH-stat method was carried out using the following formula:*DH*% = 100*BN_b_*(1/*α*)(1/*MP*)(1/*h_tot_*),
where *B* is the mL of NaOH solution needed to maintain constant pH, *N_b_* the normality of the alkaline solution, *α* the average degree of dissociation of the α-NH_2_ groups (the value for α at pH 8.5 and 25 °C is 1.16), MP is the mass (g) of protein and *h_tot_* the total number of peptide bonds in the protein substrate—the value used for whey protein concentrate being 8.8 meq/g.

### 2.3. Termination of Enzymatic Hydrolysis

Termination of enzymatic activity after defined incubation periods or on achievement of a desired %DH was performed in three different ways.

#### 2.3.1. Acid Inactivation and Determination of Residual Enzymatic Activity Effect

Hydrolysis was terminated by adding 1M HCl to achieve pH < 3. Individual samples of 20 mL were readjusted to different pH values (3.0, 6.5, 7.5, 8.5) by addition of 1M NaOH and left overnight at room temperature.

#### 2.3.2. Water Bath Heating

Glass beakers (dimension: d = 4, h = 5 cm) filled with 20 mL of hydrolysate sample were set at different pH values (4.5, 6.5, 7.5, 8.5) and heated in a water bath at 85 °C for 20 min. In order to achieve uniform temperature, the process was undertaken using a magnetic stirrer plate with Telemodul external control set at 200 rpm (Variomag^®^ Telesystem, Thermo-Fisher, Waltham, MA, USA).

#### 2.3.3. Pilot-Scale Heat Exchanger Heating

In order to achieve more rapid heating, a Microthermics UHT/HTSTL lab-25 EHVH (Raleigh, NC, USA) continuous flow heater was used. The equipment was configured to accomplish heating in two successive phases: step 1 raised the temperature of the protein hydrolysate to 60 °C; step 2 that achieved the final temperature of 85 °C with an extended holding time of 5 min, hereafter referred to as HTST-EHT. Samples (3 L) of hydrolysates were prepared and adjusted to different pH values (6.5, 7.0, 7.5, 8.0, 8.5). To avoid contamination or dilution, RO water was pumped through the process between sample runs and the initial and final 500 mL of eluate was excluded from sample collection.

Residual enzymatic activity was measured indirectly by monitoring the total protein peak on the chromatograms following analysis by high-performance liquid chromatography (HPLC).

### 2.4. Protein Profile Analysis

#### 2.4.1. High-Performance Liquid Chromatography (HPLC)

To study the evolution of the hydrolysis over time, the samples were analysed by reverse-phase (RP) HPLC using an Agilent 1200 series chromatograph (Santa Clara, CA, USA) fitted with a quaternary pump, autosampler and DAD multiple wavelength detector. Measurements were performed at 214 and 280 nm. Analysis was performed using both reverse-phase (RP) and size exclusion analysis (SEC). The RP-HPLC column, ZORBAX stableBond, 300 C-18, 4.6 × 150 mm, 5 μm, was fitted with a guard column and guard carriage (ZCG) (Agilent). The mobile phases used: Milli-Q water (Mobile phase A) and acetonitrile (Mobile phase B) (HPLC-grade ≥99.9%, Sigma-Aldrich, Hamburg, Germany) both containing 0.1% trifluoroacetic acid (TFA) (HPLC-grade provided by Sigma-Aldrich/Merck, Arklow, Ireland,). The RP-HPLC gradient used for the analysis is described in Table 1 with an equilibration (post) time of 4 min in between samples. Calibration curves were produced for individual whey proteins (β-LG, α-LA, BSA, lactoferrin) as well as WPI using known sample concentrations (range from 2.0 to 0.06 mg/mL) prepared from their respective standards (Sigma-Aldrich).

Hydrolysate samples were prepared at a concentration of 2 mg/mL by dilution in MQ water. The flow rate used was 1 mL/min with a column temperature of 35 °C, and 8 μL of sample was injected.

Molecular size distribution profiles were obtained using a size exclusion column (SEC) TSKgel G2000WXL, 7.8 × 300 mm (Tosoh Corporation, Tokyo, Japan) that was protected by a TSKgel SWXL (6 × 40 mm, 7 μm) guard column. The mobile phase in this case was a mixture of Milli-Q water (70%) and acetonitrile (30%) with an 0.1% TFA additive. A flow rate of 0.5 mL/min was maintained over a 40-min period. The sample injection volume was 20 μL at 2 mg/mL concentration. The column was calibrated with the following standards: BSA (66 kDa), a-LA (14 kDa), aprotinin (6.5 kDa), bacitracin (1.42 kDa), polypeptide (leu-Trp-Met-Arg) 604 Da and dipeptide Try-Glu (310 Da).

#### 2.4.2. Electrophoresis

Sodium dodecyl sulphate-polyacrylamide gel electrophoresis (SDS-PAGE) was performed using NuPAGE Novex bis-Tris 12-well precast gels (Invitrogen, Life Technologies Corp., Carlsbad, CA, USA), containing 4–12% polyacrylamide, prepared according to the manufacturer’s instructions under reducing and non-reducing conditions in order to study the possible aggregation of proteins using MES (2-(N-Morpholino) ethanesulfonic acid hemisodium salt) buffer. Sample volumes (10 µL) were injected into each well to achieve a protein loading of 0.65 mg/mL. A fixing solution (50% methanol and 10% acetic acid in *v*/*v*) was applied to the gels for 2 h before staining with the commercial staining solution SimplyBlue^TM^ (Thermo Scientific, Vilnius, Lithuania). Thermo Scientific’s PageRuler Unstained Low Range Protein Ladder, with its mixture of purified proteins and peptides in the size range 3.4 to 100 kDa, was used as molecular size markers.

## 3. Results and Discussion

All the Appendix A attached contain the data to support the figures and tables included and described in the text.

### 3.1. Enzymatic Hydrolysis

As described in previous work, trypsin selectively hydrolyses β-LG under specific temperature and pH conditions. In the present study, trypsin hydrolysed 100% and <33% of β-LG and α-LA (Table 1), respectively, after 3 h hydrolysis at an E:S of 1:10, which was consistent with the findings of Cheison et al. [46]. No other E:S ratio evaluated achieved 100% hydrolysis of β-LG, irrespective of the incubation times studied; however, ~90% hydrolysis could be achieved using an E:S of 1:30 and 1:50 within a 4-h incubation period. Extending the incubation period to 6 h using an E:S of 1:50 only marginally increased β-LG hydrolysis >90%.

Lowering E:S from 1:10 to 1:30 reduced the extent of hydrolysis of β-LG and α-LA to 86.5 ± 1.06% and 11.83 ± 0.9%, respectively, during a 3-h incubation. A further decrease in E:S to 1:50 marginally reduced the extent of β-LG hydrolysis to 81.04 ± 0.81%, but with a beneficial effect in terms of limiting the extent of α-LA hydrolysis, which was reduced to <5%. An E:S of 1:100 during a similar 3-h incubation period resulted in 53.15 ± 2.87% hydrolysis of β-LG, and this value could be extended further to 77.54 ± 1.01% during a 7-h incubation. While 86% hydrolysis of β-LG may be achieved during a shorter duration (1 h) of incubation using an E:S of 1:10, its more intense hydrolytic activity resulted in greater hydrolysis of α-LA (25%).

The evolution of hydrolysis at all E:S studied (based on %DH) and the change in composition of whey protein is represented in Figure 1 and Table 2, respectively. Within the DH range 4.5–6.5%, trypsin hydrolysed β-LG to >90% while α-LA remained almost unhydrolysed (<10%), thus indicating key parameters that could be employed for selective recovery of the latter fraction in a near-intact form. Above DH 6.5%, the extent of hydrolysis of α-LA increased.

Thus, the protein hydrolysis protocols chosen for further experimental studies involved an E:S 1:50 and an incubation time of 3 h based on targeted protein hydrolysis levels of ~5% α-LA and ~80% β-LG. These conditions should equate to an incubation time of moderate duration and a relatively low enzyme cost.

### 3.2. Inactivation Assays

#### 3.2.1. Acid Inactivation

There was little evidence of trypsin activity at pH < 3. Enzyme reactivation, however, was observed once the pH was restored stepwise to pH 8.5. The effect after overnight storage of the hydrolysate at room temperature (~20 °C) on the further degradation of α-LA and β-LG, as determined by RP-HPLC, is represented in Figure 2. Between 25 and 30% of the remaining α-LA was hydrolysed within the range pH 5.5–8.5 under these conditions (Figure 2), while those samples adjusted to pH < 3 were not further hydrolysed following overnight storage. It was also notable that α-LA in the samples held within the pH range 5.5–6.5 was preferentially hydrolysed by trypsin relative to β-LG. However, that trend was reversed on pH adjustment to 7.5, resulting in a more extensive hydrolysis of β-LG by trypsin on storage at near-neutral pH conditions. Finally, as expected, trypsin preferentially increased the hydrolysis of β-LG at pH 8.5, as was reported by Cheison et al. [43].

Thus, the termination of enzymatic reactions by lowering the pH is challenging as any subsequent pH alteration can lead to enzyme reactivation and further non-targeted hydrolysis. In any case, the limitation of having to maintain the hydrolysate pH within an acidic pH range restricts further analysis and end-product application.

#### 3.2.2. Heat Inactivation: Water Bath

At the laboratory scale, a water bath is the most commonly used method for enzyme inactivation. In order to reduce the heating time, it is necessary to increase the temperature and minimise the sample volume while stirring. However, laboratory heating proved to be ineffective at controlling the trypsin hydrolysis of α-LA during the inactivation process, according to the data obtained from the protein fraction in the RP-HPLC chromatograms at all pH values (Figure 3). Figure 3 also shows that trypsin was still selectively hydrolysing β-LG during heat treatment in the pH range 6.5–8.5, thus resulting in an increased number of peptides produced, although the enzyme was inactivated by the end of the heating. In the case of the samples at pH 7.5 and 8.5, both α-LA and β-LG were completely hydrolysed, thus rendering it impossible to recover α-LA. At pH 6.5 the hydrolytic effect was less pronounced, although much of the intact α-LA and β-LG had disappeared. Just as in the case of the sample held at pH 4.5, the remaining β-LG was not hydrolysed, yet α-LA was not detected.

This loss of enzyme selectivity, together with continuing hydrolysis, occurred during heat inactivation of trypsin at a relatively high denaturation temperature of 85 °C for 20 min. In order to attain the target inactivation temperature of 85 °C during heating from 25 °C, it was necessary during the ramping-up period to traverse the region of optimum temperature for trypsin activity, i.e., between 3 °C and 55 °C. As a result, the preferential enzyme selectivity observed at 25 °C was lost during this heat inactivation treatment, thus negating the opportunity to recover α-LA.

#### 3.2.3. Heat Inactivation: Heat Exchanger

In order to improve the thermal denaturation kinetics of trypsin in a whey protein hydrolysate medium, a Microthermics^®^ continuous HTST-EHT heat exchanger was used to rapidly increase the temperature during the heating step, thus minimising the possibility of continuing hydrolysis during transition through the enzyme’s operating temperature range. Figure 4 shows the effect of HTST-EHT heat treatment on samples treated at different pH values (6.5, 7, 7.5, 8, 8.5) on the molecular mass profiles when analysed by SEC-HPLC. The results were plotted in conjunction with the elution time of the protein standards: α-LA (14 kDa), β-LG (36 kDa) and BSA (63 kDa) (Figure 4). Surprisingly, there was little evidence of either of these whey protein fractions in the chromatograms of the HTST-EHT heat-inactivated hydrolysates, irrespective of the heating pH used. However, aggregation was evident in all HTST-EHT-heated hydrolysate samples. These aggregates (>66 kDa) are considered to be as a result of the high temperature of 85 °C used to inactivate trypsin, during which protein/peptide aggregate complexes are formed by non-covalent interactions and disulphide aggregation. This development obstructed further quantification and detailed study of the targeted hydrolysates [46,47].

SDS-PAGE under reducing and non-reducing conditions was also used to analyse the effect of heat treatment on the protein hydrolysate. Under reducing conditions (Figure 5A), the aggregates were not evident. At pH 6.5, 7.0 and 7.5 the intensity of the bands of β-LG and α-LA was lower, indicating that both proteins underwent some hydrolysis consistent with the optimum pH for trypsin activity and, thus, affecting the selectivity for α-LA. In the case of samples at pH 8.0 (Lane 5) and 8.5 (Lane 6), both proteins remained intact, confirming the achievement of the desired effect of the selective hydrolysis protocol. Under non-reducing conditions (Figure 5B), it was not possible to detect any β-LG or α-LA bands in lanes 2–6, representing the pH range 6.5–8.5, although there was a faint band representing BSA in lanes 4–7, which coincided with a smaller variation in pH 7.5–8.5. However, under reducing conditions (Figure 5A), bands representing β-LG and α-LA are clearly evident in the pH 8 and 8.5 (lanes 5 and 6) samples, along with BSA (66 kDa). Such a difference generated by SDS-PAGE analysis confirms the occurrence of protein‒peptide aggregation produced by HTST-EHT heat treatment, and the reducing conditions necessary to achieve aggregate breakdown.

HTST-EHT continuous thermal processes such as those used industrially would, therefore, be expected to minimise the extent of intact α-LA loss compared with batch heating, e.g., the water bath method as described in this study, due to the lag time taken to reach the inactivation temperature. However, aggregate formation induced during HTST-EHT heat inactivation of a trypsin-containing whey protein hydrolysate makes it practically impossible to recover unhydrolysed α-LA.

## 4. Conclusions

In the context of future process scale-up and the necessity of controlling the costs of enzyme addition, the optimum E:S was determined to be 1:50 with extension of incubation time from 3 to 7 h in order to selectively achieve 100% hydrolysis of β-LG with minimal degradation of α-LA. Further experiments with an E:S of 1:100 may also be of interest in order to determine the total time needed for 100% hydrolysis of β-LG, while at the same time considering the economic consequences of running hydrolysis over a protracted period. All traditional methods employed to terminate enzymatic activity proved effective, but had a considerable negative impact on the target protein, i.e., α-LA. Acidification to <pH 3 limits possible further application of the hydrolysate, as all functionality and formulation could not be conducted at such an acidic pH. Heat treatment by means of continuous HTST-EHT heating was effective at limiting further protein hydrolysis, with trypsin selectivity affected by the pH—at neutral pH conditions, the selectivity of hydrolysis was compromised and intact α-LA began to be digested, while selectivity for maintaining intact α-LA was not affected at a higher pH (pH > 8). However, heat-induced protein/peptide aggregate formation makes it practically impossible to recover unhydrolysed α-LA intact. For these reasons, continuous HTST-EHT heating was identified as unsuitable for terminating tryptic-led WPI hydrolysis, especially when the process is aimed at limiting the breakdown of intact α-LA. Further studies are required to determine whether the aggregates formed may be broken down subsequently in order to facilitate the recovery of intact α-LA. Other processing options should be investigated for managing and controlling enzyme activity in order to facilitate the downstream recovery of intact α-LA [48,49,50].

## Figures and Tables

**Figure 1 foods-08-00367-f001:**
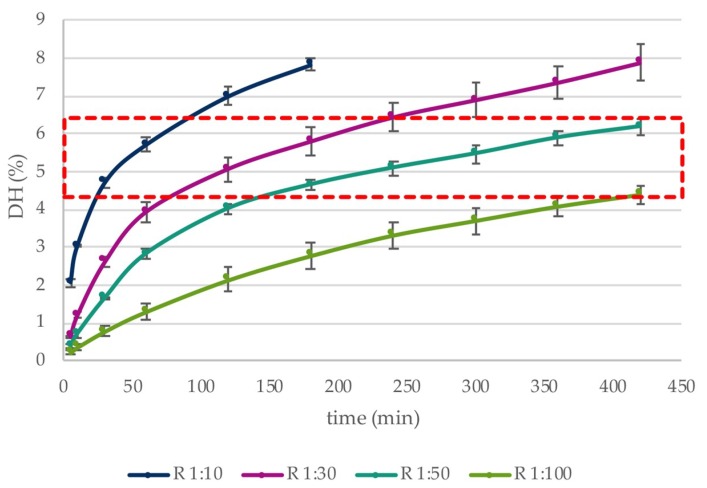
The effect of a targeted hydrolysis protocol (pH 8.5, 25 °C) on the % degree of hydrolysis (%DH) vs. incubation time (min) of whey protein isolate using trypsin at different enzyme:substrate ratios (E:S 1:10, 1:30, 1:50, 1:100). All trials were conducted in triplicate. Values represent mean ± standard deviation.

**Figure 2 foods-08-00367-f002:**
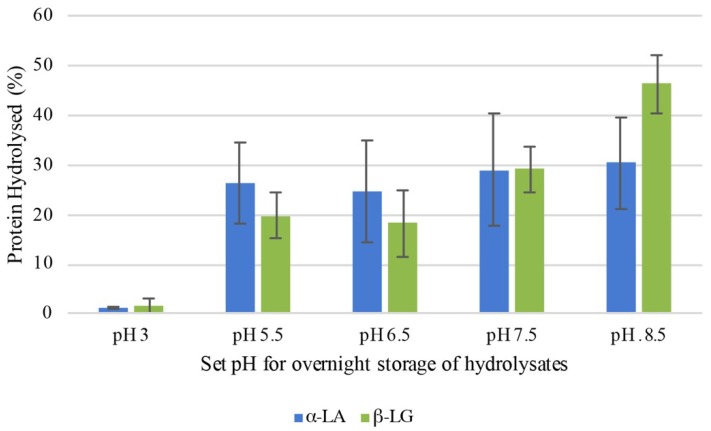
Percentage change in hydrolysis of α-lactalbumin (α-LA) and β-lactoglobulin (β-LG) following overnight storage at room temperature (20 °C) and different pH values (3.0, 5.5, 6.5, 7.5, 8.5) of a tryptic whey protein isolate hydrolysate, analysed in triplicate.

**Figure 3 foods-08-00367-f003:**
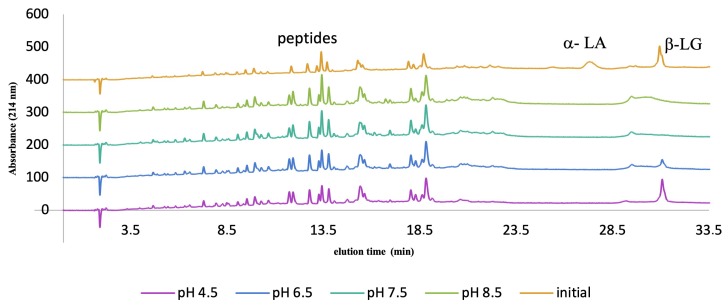
Reverse-phase high-performance liquid chromatography profiles of a tryptic hydrolysate of whey protein isolate at different pH values (4.5, 6.5, 7.5, and 8.5) following water bath heating to 85 °C for 20 min for the purpose of trypsin inactivation. α-LA(α-lactalbumin) β-LG (β-lactoglobulin).

**Figure 4 foods-08-00367-f004:**
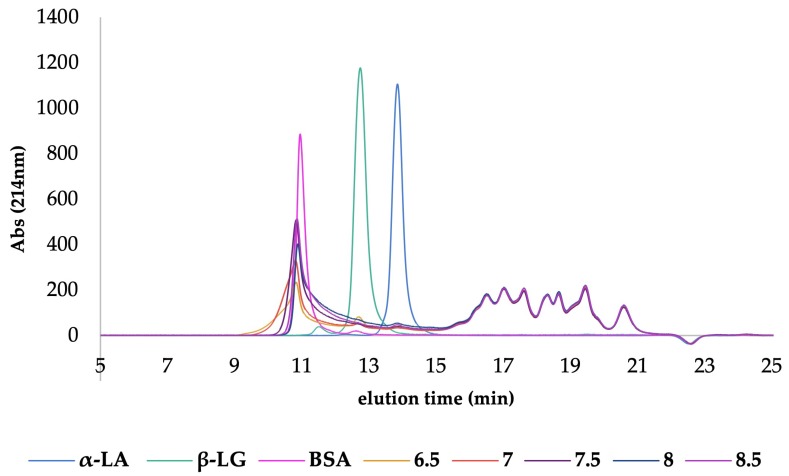
High-performance liquid chromatography‒size exclusion (HPLC-SEC) profiles, representing the effect of continuous heat and holding (HTST-EHT) treatment using a heat exchanger on selectively hydrolysed whey protein isolate samples held at different pH values. Protein standards: α-lactalbumin (α-LA), β-lactoglobulin (β-LG), and bovine serum albumin (BSA).

**Figure 5 foods-08-00367-f005:**
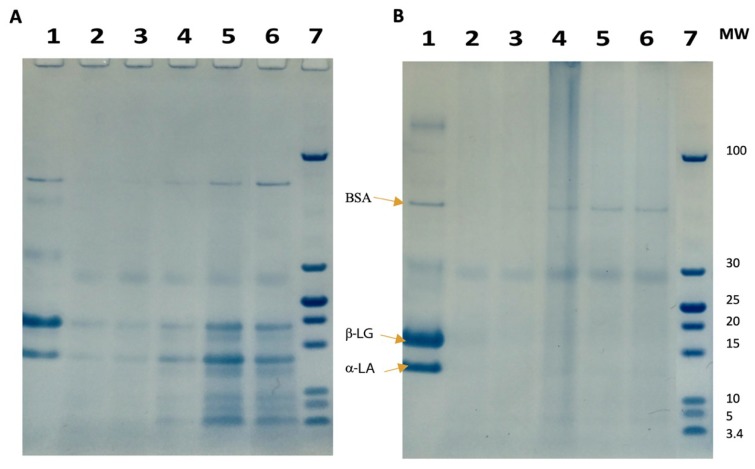
Sodium dodecyl sulphate polyacrylamide electrophoresis (SDS-PAGE) of tryptic hydrolysates of whey protein isolate following heating to 85 °C for 5 min at different pH values, using a heat exchanger, under (**A**) reducing conditions (1) WPI; (2) pH 6.5; (3) pH 7; (4) pH 7.5; (5) pH 8; (6) pH 8.5; (7) molecular weight marker; (**B**) SDS-PAGE under non-reducing conditions, (1) WPI, (2) pH 6.5; (3) pH 7; (4) pH 7.5; (5) pH 8; (6) pH 8.5; (7) molecular weight markers.

**Table 1 foods-08-00367-t001:** High-performance liquid chromatography‒reverse-phase solvent gradients used at various time points for protein‒peptide separation and quantification.

t (min)	% Acetonitrile
0	3.0
20	38.4
26	38.4
28	44.0
31	52.0
33	90.0
37	90.0
38	3.0

**Table 2 foods-08-00367-t002:** Comparison of overall degree of hydrolysis (%DH) and extent of hydrolysis (%) of β-lactoglobulin and α-lactalbumin at different enzyme: Substrate ratios (E:S 1:10, 1:30, 1:50 and 1:100) and incubation times (1 to 7 h) during the targeted hydrolysis of whey protein isolate with trypsin during incubation at 25 °C and pH 8.5. Mean values and standard deviation (*n* = 3) were derived from the results of triplicate trials.

	E:S 1:10			E:S 1:30			E:S 1:50			E:S 1:100		
Time (h)	%DH	α-LA (%)	β-LG (%)	%DH	α-LA (%)	β-LG (%)	%DH	α-LA (%)	β-LG (%)	%DH	α-LA (%)	β-LG (%)
1	5.71 ± 0.19	26.84 ± 2.61	86.46 ± 1.87	3.93 ± 0.26	7.43 ± 1.83	70.36 ± 3.10	2.82 ± 0.14	0.05 ± 5.79	58.67 ± 1.51	1.30 ± 0.24	2.39 ± 2.93	30.63 ± 1.58
2	7.00 ± 0.22	31.02 ± 2.15	97.49 ± 0.76	5.07 ± 0.32	9.91 ± 3.27	80.63 ± 0.57	4.01 ± 0.13	0.84 ± 2.34	73.72 ± 1.67	2.14 ± 0.32	2.54 ± 2.23	47.34 ± 1.53
3	7.83 ± 0.17	33.18 ± 1.67	100.00	5.79 ± 0.37	11.83 ± 0.93	86.55 ± 1.06	4.64 ± 0.15	1.51 ± 5.20	81.04 ± 0.81	2.78 ± 0.35	3.52 ± 2.53	53.15 ± 2.87
4				6.43 ± 0.38	11.59 ± 0.93	90.54 ± 1.11	5.08 ± 0.20	3.83 ± 6.92	86.19 ± 0.19	3.32 ± 0.36	3.47 ± 4.99	63.90 ± 0.84
5				6.88 ± 0.44	9.82 ± 2.88	93.14 ± 0.40	5.46 ± 0.23	4.11 ± 3.44	88.32 ± 0.44	3.70 ± 0.34	3.00 ± 1.82	69.81 ± 2.19
6				7.34 ± 0.43	11.06 ± 4.43	95.38 ± 0.60	5.88 ± 0.19	4.43 ± 3.47	91.71 ± 0.45	4.08 ± 0.26	3.13 ± 2.28	73.27 ± 1.35
7				7.86 ± 0.48	14.10 ± 3.31	97.22 ± 0.50	6.17 ± 0.20	4.68 ± 2.88	93.85 ± 0.31	4.39 ± 0.23	4.48 ± 3.22	77.54 ± 1.01

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
