# Peer review of "Exploring the Use of a Modified High-Temperature, Short-Time Continuous Heat Exchanger with Extended Holding Time (HTST-EHT) for Thermal Inactivation of Trypsin Following Selective Enzymatic Hydrolysis of the β-Lactoglobulin Fraction in Whey Protein Isolate"

_foods, 2019, doi:10.3390/foods8090367_

Round 1
Reviewer 1 Report
My overall judgement is "Minor revision", due to some considerable technical issues, mainly.
- Regarding continuous HTST for enzyme inactivation: In the introduction the expectation should be formulated that this treatment could well lead to losses of peptides due to aggregation/denaturation. If this was not expected by the authors they should formulate a clear hypothesis related to the kinetics of enzyme inactivation being much faster than the kinetics of peptide aggregation (which then later could be disconfirmed by the results).
- Regarding their conclusions and mentioning the concept of EMBR the authors are recommended to consider the publications by Cheison in 2006 and 2007 in J. Membrane Sci., Int. Dairy J. and J. Food Eng.
Technical aspects
- There should be blanks in equations before and after the =
- The same applies for blanks between the numbers and related SI units, including %
- In the section of Termination of enzymatic hydrolysis, 1. Acid hydrolysis ist says that "Hydrolysis was determinate..." . It should read "... was terminated"
- Line 52 (the front part does not have line numbers): 55 °
- Citations should be consistently cited as [number], e.g. also Lisak et al. (2013) --> Lisak et al. [41]
- List of references: In most citations the details regarding volume, page numbers etc are missing. Also, the citations should be consistent in style (e.g. Journal names abbreviated or not, author names not fully in capital letters). All details should be checked against style guide for authors.
Author Response
In first place I would like to thank you for your time with the revision. We really appreciate all the comments you did, I attach the word file that I hope will clear all the mistakes and doubts. Due to the suggestion of other reviewers some parts of the text have been changed.
Regarding to your specific comments:
- Regarding continuous HTST for enzyme inactivation: In the introduction the expectation should be formulated that this treatment could well lead to losses of peptides due to aggregation/denaturation. If this was not expected by the authors they should formulate a clear hypothesis related to the kinetics of enzyme inactivation being much faster than the kinetics of peptide aggregation (which then later could be disconfirmed by the results).
RESPONSE: We appreciate the Reviewer’s comment which are correct, aggregation protein- peptides and denaturalization was an effect expected. The last paragraph of introduction has been modified to include it.
Lines 150-154 (with track changes active):
94-99 (changes accepted)
“In this study, a continuous high temperature, short time heater (HTST) with extended holding time (EHT) based thermal treatment was assessed as a means of inactivating trypsin-based selective hydrolysis of β-lg in a WPI substrate dispersion while quantifying the formation of undesirable side reactions associated with protein denaturation and protein-peptide aggregation. An ultimate objective being to minimise possible aggregate formation that could interfere with subsequent research efforts aimed at producing an a-lac enriched product from the targeted WPI hydrolysis. Furthermore, since original research [43]using trypsin was conducted at a relatively high E:S of 1:10, a further objective was to evaluate a reduced E:S down to 1:100 on the incubation conditions necessary to maximise β-lg digestion while minimising the digestion of a-lac. “
- Regarding their conclusions and mentioning the concept of EMBR the authors are recommended to consider the publications by Cheison in 2006 and 2007 in J. Membrane Sci., Int. Dairy J. and J. Food Eng.
RESPONSE: Due to possible confusion with the final conclusions, that have been noted by other reviewers, it has been decided to delete the reference specific reference to EMBR, however we appreciate the references suggested which will be considered by the project team as we address the use of EMBR to improve the selective hydrolysis.
Deletion of Lines 323-326:
“One such option is the use of an enzymatic membrane bioreactor (EMBR) which will be examined during future studies because of its facility to continually separate an a-lac rich stream from a progressively hydrolysing b-lg permeate phase throughout the process, this method have been previously successfully employed in enzymatic hydrolysates production as method to remove the enzyme”
Modified text, lines 489-614(with track changes active)
lines 333-336 (changes accepted)
“Further studies are required to determine whether the aggregates formed may be broken down subsequently in order to facilitate recovery of intact a-lac. Other processing options should be investigated into managing and controlling enzyme activity in order to facilitate downstream recovery of intact a-lac[49–51].”
Technical aspects
- There should be blanks in equations before and after the =
- The same applies for blanks between the numbers and related SI units, including %
- In the section of Termination of enzymatic hydrolysis, 1. Acid hydrolysis ist says that "Hydrolysis was determinate..." . It should read "... was terminated"
- Line 52 (the front part does not have line numbers): 55 °
- Citations should be consistently cited as [number], e.g. also Lisak et al. (2013) --> Lisak et al. [41]
- List of references: In most citations the details regarding volume, page numbers etc are missing. Also, the citations should be consistent in style (e.g. Journal names abbreviated or not, author names not fully in capital letters). All details should be checked against style guide for authors.
RESPONSE: We appreciated the technical aspects detected (specially references), all have been now corrected.

Reviewer 2 Report
Dear Editor,
the authors described the better condition of E/S, pH and temperature to hydrolyzed only beta-LG leaving intact the alfa-LA through the utilize of tripsin. Moreover, they consider also the conditions to inactivate the enzyme proposing as a new method the HTST treatment. They concluded that neither the use of decreasing of pH nor heat treatment are satisfaying to recover completely the functionality of protein and peptides.
Major Comments,
1. the use of heat treatment (85 °C for 5 min), which is not a HTST processes (about 72 °C for 10 sec), even knowing that the denaturation of whey proteins start at 65-70 °C.
2. the reason of recover of native a-LA is diluted in the text thereby “the traces are lost”. It is not clear what can be the use of hydrolysed whey containing native a-LA: formulation of infant powder milks? Formulation of food, but what? What is the advantage of native a-LA in proteolyzed whey? As a matter of fact, the paper is focused mainly on the finding of better condition for hydrolysing whey proteins, as if it a technical note. In my opinion the authors should spend some more word as concerns the role of native a-LA in food formulations.
3. In the conclusions the authors verified that heat treatments were not appropriated to obtain native a-LA and proposed: “the use of an enzymatic membrane bioreactor (EMBR) which will be examined during future studies because of its facility to continually separate an a-lac rich stream from a progressively hydrolysing b-lg permeate phase throughout the process”. Therefore, overall considerations, I wonder what the authors want to prove in this paper?
4. Finally, it is well known that some functional characteristic, such as foaming, thickening and emulsification, improve with protein denaturation.
Minor comment,
I think that RP-HPLC, which the acronym of reversed phase-high performance liquid chromatography, is more correct than HPLC RP.
Author Response
In first place I would like to thank you your time with the revision. We really appreciate all the comments you did, I attach the word file that I hope will clear all the mistakes and doubts. Due to the suggestion of other reviewers some parts of the text have been changed.
Regarding to your specific comments:
Major Comments,
the use of heat treatment (85 °C for 5 min), which is not a HTST processes (about 72 °C for 10 sec), even knowing that the denaturation of whey proteins starts at 65-70 °C.
REPONSE: We appreciate the Reviewer’s comment which is technically correct. In order to reflect the process conditions used which involved modification of ‘HTST heating’ with extended holding time (EHT) we have amended all reference to HTST throughout the manuscript with the term ‘HTST-EHT heating’.
the reason of recover of native a-LA is diluted in the text thereby “the traces are lost”. It is not clear what can be the use of hydrolysed whey containing native a-LA: formulation of infant powder milks? Formulation of food, but what? What is the advantage of native a-LA in proteolyzed whey? As a matter of fact, the paper is focused mainly on the finding of better condition for hydrolysing whey proteins, as if it a technical note. In my opinion the authors should spend some more word as concerns the role of native a-LA in food formulations.
REPONSE:The following texts have been inserted to clarify the purpose of focusing on a-lac from an infant nutrition perspective (Lines 117-124), while the final paragraph (Lines 149-154) updates the hypothesis of pursuing a successful targeted hydrolysis on the one hand that is aimed at protecting a-lac intact, and the subsequent complication presented during post hydrolysis step(s) when heat-induced enzyme inactivation can frustrate the efforts of a successful targeted hydrolysis as a result of side reactions.
Lines 117-124 (With track changes active):
75-82 (changes accepted)
“In the case of WPI, a selective hydrolysis of β-lg may reduce the risk of allergy and at the same time enables the generation of a co-product enriched in a-lac – the predominant whey protein fraction in human milk which is a key nutritional source of essential amino acids in sufficient concentration to promote infant development. Supplementation of infant formula with a-lac helps to lower the total protein content of formula to resemble that of human milk. In addition, a-lac possesses a wide range of biological activities such asabsorption of minerals, antibacterial activity, antioxidant activity, immunomodulatory effects and antitumor activity which promote the health of the neonate [41,42]”
Lines (149-154)(With track changes active):
94-99 (changes accepted)
“In this study, a continuous high temperature, short time heater (HTST) with extended holding time (EHT) based thermal treatment was assessed as a means of inactivating trypsin-based selective hydrolysis of β-lg in a WPI substrate dispersion while quantifying the formation of undesirable side reactions associated with protein denaturation and protein-peptide aggregation. An ultimate objective being to minimise possible aggregate formation that could interfere with subsequent research efforts aimed at producing an a-lac enriched product from the targeted WPI hydrolysis.”.
In the conclusions the authors verified that heat treatments were not appropriated to obtain native a-LA and proposed: “the use of an enzymatic membrane bioreactor (EMBR) which will be examined during future studies because of its facility to continually separate an a-lac rich stream from a progressively hydrolysing b-lg permeate phase throughout the process”. Therefore, overall considerations, I wonder what the authors want to prove in this paper?
RESPONSE: The following lines have been deleted from the Conclusion as they do not relate to the subject of the paper and could confuse:
Deletion of Lines 323-326:
“One such option is the use of an enzymatic membrane bioreactor (EMBR) which will be examined during future studies because of its facility to continually separate an a-lac rich stream from a progressively hydrolysing b-lg permeate phase throughout the process, this method have been previously successfully employed in enzymatic hydrolysates production as method to remove the enzyme”
Lines 430-489 (With track changes active):
324-333 (with changes accepted)
RESPONSE: The text in the following lines have been edited to strengthen the study’s findings
“Acidification to 8) was not affected – heat-induced protein/peptide aggregate formation makes practically impossible to recover targeted un-hydrolysed intact a-lac. For these reasons, continuous HTST-EHT heating was identified during this study as unsuitable for terminating tryptic-led WPI hydrolysis especially when the process is aimed at limiting the breakdown of intact a-lac. Further studies are required to determine whether the aggregates formed may be broken down subsequently in order to facilitate recovery of intact a-lac. Other processing options should be investigated into managing and controlling enzyme activity in order to facilitate downstream recovery of intact a-lac[49–51].”
Finally, it is well known that some functional characteristic, such as foaming, thickening and emulsification, improve with protein denaturation.
RESPONSE: Certainly, many bio-technological properties are improved after heat-treatment of proteins, further studies of our group have been focused in the study of these properties and the ones of the samples obtained after EMBR, however, the present work was more focus in optimization of hydrolysis conditions and methods of inactivation, as initial step of hydrolysate enriched in a-lac for further applications.
Kind regards

Round 2
Reviewer 2 Report
Dear Editor,
The authors done a good work, however, I would suggest to the authors to use the common nomenclature, as follows:
change α-lac to α-LA
change β-lg to β-LG
Author Response
In first place we would like to thank you for your time.
The changes that you suggested are now done in the text. All the abbreviation for whey proteins have been changed in the text and figures as well.
Please see the attached document with the changes
Kind regards.
